# Microstructure and Nanoindentation Behavior of Ti$_{40}$Zr$_{40}$Ni$_{20}$ Quasicrystal Alloy by Casting and Rapid Solidification

Junli Hou [ID], Zhong Yang *, Hongbo Duan, Yiyi Feng, Yongchun Guo and Jianping Li

School of Materials and Chemical Engineering, Xi'an Technological University, Xi'an 710021, China; houjunli@st.xatu.edu.cn (J.H.); duan_hbxy@163.com (H.D.); fengyiyi@st.xatu.edu.cn (Y.F.); 18706752776@163.com (Y.G.); jpli01@xatu.edu.cn (J.L.)
* Correspondence: yz750925@163.com

**Abstract:** A Ti$_{40}$Zr$_{40}$Ni$_{20}$ quasicrystal (QCs) rod and ribbons were prepared by conventional casting and rapid solidification. The X-ray diffractometry (XRD), scanning electron microscopy (SEM), transmission electron microscopy (TEM) and differential scanning calorimeter (DSC) techniques were used to investigate the microtissue, phase composition, and solidification features of the samples; the nano-indentation test was carried out at room temperature. The results show that a mixture of the α-Ti(Zr) phase and the icosahedral quasicrystal (I-phase) was formed in the Ti$_{40}$Zr$_{40}$Ni$_{20}$ rod; the microstructure of Ti$_{40}$Zr$_{40}$Ni$_{20}$ ribbons mainly consisted of the I-phase. The solidification mechanism of the I-phase was different in the two alloys. The I-phase in the quasicrystalline rod was formed by packet reaction while in the ribbons it was generated directly from the liquid. At room temperature, both samples had relatively high hardness and elastic modulus; the elastic modulus of the ribbons is 76 GPa, higher than the 45 GPa of the rod. The hardness of the ribbons was more than twice that of the rod.

**Keywords:** Ti-Zr-Ni alloy; quasicrystal; microstructure; nanoindentation

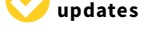



## 1. Introduction

Quasicrystals (QCs), a particular solid-state ordered phase, have a quasi-periodic translational order and amorphous symmetry [1–4]. Due to their distinctive atomic structure, QCs possess many desirable properties [2,3] such as low thermal conductivity [5,6], low surface energy [7], low friction coefficients [8,9], high hardness and high wear resistance [10,11], etc. These critical properties promote the application of QCs as thin films [10,12], coatings [13,14], and reinforced particles in many fields such as solar energy [13–18], automobile manufacturing, aerospace technologies, etc. Together with superconductors, quasicrystals are regarded as the two most significant advances in condensed matter physics in the 1980s, and are still regarded as a frontier discipline in this field [19]. Ti-based QCs containing a thermodynamically stable icosahedral phase (I-phase) have attractive properties such as high strength at elevated temperatures and low friction coefficients [20,21]. Furthermore, Ti-based QCs not only show the original characteristics of QCs but also show superior hydrogen storage capacity [21–24]. Therefore, as one of the hydrogen storage and negative electrode materials for nickel/metal hydride (Ni/MH) secondary batteries, Ti-based QCs have potential applications in both power generation and in environmental protection. However, there is little research on Ti-based QCs as the reinforcing phase of structural materials. Among Ti-based QCs, the TiZrNiQCs, first reported by Molokanov et al. [25], are strong candidates for practical applications due to their characteristics of easy formation, relatively large grain size and excellent performance, as mentioned in [23]. While the I-phase is thermodynamically stable [26], the I-phase material in the Ti-Zr-Ni system is hard to fabricate through conventional casting. Nevertheless, Qiang successfully prepared high-quality block quasicrystalline alloy with self-made split water-cooled cooper mold suction casting equipment [27]. Certainly, rapid solidification is the most common method

to prepare a single I-phase [28]. Due to size problems with the prepared sample, it is difficult to characterize the hardness and elastic modulus of TiZrNiQCs.

In this paper, $Ti_{40}Zr_{40}Ni_{20}$ QCs were prepared by the Cu-mold cooled blow casting and rapid solidification technique. In addition to thermal stability, the effect of structure on elastic modulus as well as on the hardness indexes of $Ti_{40}Zr_{40}Ni_{20}$ QCs alloys was investigated.

## 2. Experimental Procedure

### 2.1. Arc Melting

The main raw materials (Ti 99.9, Zr 99.5 and Ni 99.9 mass% purity) were obtained from the following commercial sources: Beijing Yanbang New Material Technology Co. LTD (Bejing, China). Alloy ingots of $Ti_{40}Zr_{40}Ni_{20}$ (at%) were prepared by arc-melting under an argon atmosphere (Purity > 89.9%). The working voltage and current of the arc melting furnace were 70 V and 600 Amps. To ensure the uniformity of the ingots, each ingot was refined at least three times.

### 2.2. Copper Blow Casting

From these ingots, the $Ti_{40}Zr_{40}Ni_{20}$ alloy rod with Φ4 mm was prepared using copper mold blow-casting equipment (Shenyang Haosiduo New Material Preparation Co., LTD, Shenyang, China).

The small pieces (10 g) of the chopped alloy ingot were put into a quartz tube with a nozzle diameter of 0.5 mm, and the heating power was gradually increased to 28 kW, which was maintained for 1 min and then reduced to 20 kW for the blow casting experiment under an argon atmosphere (Purity > 89.9%). Then, the pressure regulator valve was opened and the pressure difference used to spray the molten alloy liquid into the cylindrical cavity of the water-cooled copper mold to solidify. The cooling rate near the bottom of the copper mold was higher than that at the top, with a maximum over $10^3$ K/s. The rod samples were subjected to annealing experiments in vacuum annealing furnaces.

### 2.3. Rapid Quenching

Metallic ribbons with 40–110 μm thickness and 2–4 cm in length were prepared by rapid solidification of the melt on a single copper roller with 2500 rotations per minute.

The alloy block (10 g) was placed into a quartz pipe with the nozzle shape of a flat port. In an environment of high pure argon protection (Purity > 89.9%), an alloy block was melted into liquid by magnetic induction smelting heating. The heating power was gradually increased to 33 kW, which was maintained for 1 min and then reduced to 30 kW for the rapid quenching experiment. The molten liquid was sprayed onto the surface of a rapidly rolling copper roller with a diameter of 200 mm using gas pressure differences. Because the cooling speed can reach an order of $10^4$–$10^6$ K/s, this process required precise timing.

### 2.4. Analysis and Detection Methods

The microstructure and rotational symmetry of the rod and ribbon samples were examined by X-ray diffractometry (XRD-6000, Shimadzu, Japan) with monochromatic Cu Kα radiation (λ = 0.1542 nm) and a JEOL JEM-2010 transmission electron microscopy (TEM, JEOL, Tokyo Metropolitan, Japan). Specimens for TEM observation were prepared by the ion polishing technique. The morphology and composition of the samples were characterized using scanning electron microscopy (SEM, TESCAN, ORSAY HOLDING, Brno, Czech Republic). To analyze the components of the specimen, an energy dispersive spectrometer (EDS, TESCAN, ORSAY HOLDING, Brno, Czech Republic) was used. Hydrofluoric acid and nitric acid aqueous solution (10 mL water + 8 mL $HNO_3$ + 3 mL HF) was used as an etchant on the polished samples. The phase transformation was investigated by a high-temperature differential scanning calorimeter (TGA/DSC1, METTLER,

Toledo, Switzerland). The DSC curves were obtained by placing about 60 mg of sample in an open ceramic crucible at a heating rate of 10 K/min in an argon flow of 50 mL/min.

The nanoindentation test was conducted using the Bruker Hysitron TI Premier system for samples with load levels ranging from 1 mN to 10 mN. The measured hardness and modulus were the average values of eight points.

## 3. Results and Discussion

### 3.1. Structure Characterization and Morphology

The X-ray diffraction patterns of Ti-40at.%Zr-20at.%Ni alloy prepared by blow-casting and melt-spinning tools are shown in Figure 1. A hump-like a region appeared in all the XRD patterns, which are a result of the XRD sample holder. It can be seen from Figure 1a that the XRD pattern reveals the presence of both icosahedral quasicrystals (I-phase indexed according to Cahn scheme [29]) and $\alpha$-solid solution phase. The sharp peaks of the icosahedral quasicrystalline phase are around 35–39°, and the strongest diffraction peak of the $\alpha$-solid solution phase are identified at 34°, 40° and 41°. The weak peak of $\alpha$-solid solution phase in the XRD measurements (Figure 1a) might be attributed to small grain size or its low content in the alloys. The XRD pattern of the ribbons (as shown in Figure 1b) has a peak index with only the I-phase. The lattice constant of the I-phase of both samples was to be 0.508 nm, conforming to the value quasilattice of this alloy [27], and the $\alpha$-solid solution phase was calculated as a = 0.311 nm and c = 0.493 nm, respectively.

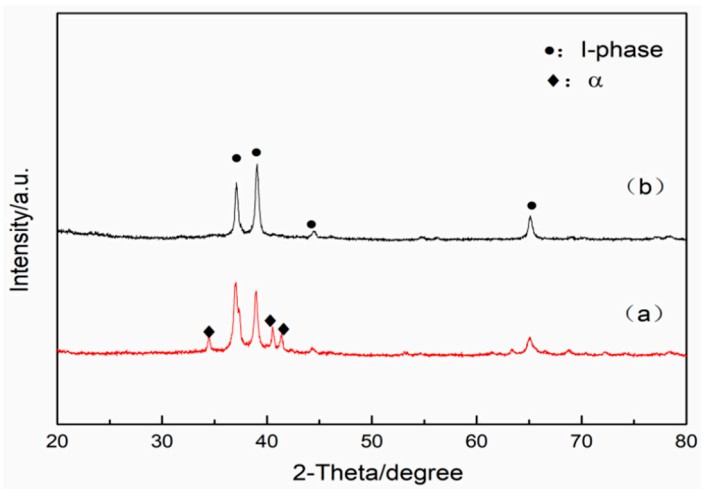

**Figure 1.** XRD patterns of (**a**) The $Ti_{40}Zr_{40}Ni_{20}$ rod and (**b**) The $Ti_{40}Zr_{40}Ni_{20}$ ribbons.

The SEM images and energy spectrum analysis of the $Ti_{40}Zr_{40}Ni_{20}$ rod are shown in Figure 2a,c,d. The energy spectrum components of all points are similar, and their average values are shown in the interpolation table in Figure 2c,d. The microstructure of the rod is characterized by the dendritic primary crystal phase (A) distributed in the crystal boundary and block I-phase (B) grains of 4–17 μm in size (Figure 2a). Confirmed by EDS quantitative studies, results indicate that the neighbouring two phases have different chemical compositions. Among the primary crystals there is an $\alpha$-solid solution phase with the composition of $Ti_{37\pm0.2}Zr_{34\pm0.4}Ni_{28\pm0.1}$ and $Ti_{39\pm0.4}Zr_{31\pm0.4}Ni_{29\pm0.1}$ for the I-phase, supporting the X-ray diffraction data. This implies that the dendrite $\alpha$-solid solution phase in the rod alloy had a high content of Ni. Furthermore, scanning electron micrographs of the ribbons, as shown in Figure 2b, have smaller microtissue, with the composition given below the article.

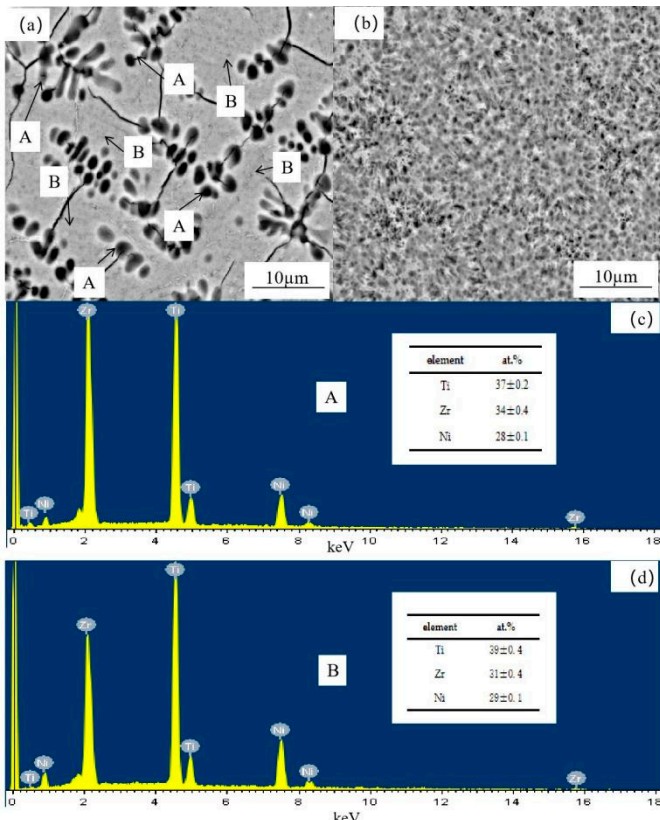

**Figure 2.** SEM morphology of: (**a**) The $Ti_{40}Zr_{40}Ni_{20}$ rod; (**b**) The $Ti_{40}Zr_{40}Ni_{20}$ ribbon. The EDS spectrum of the $Ti_{40}Zr_{40}Ni_{20}$ rod: (**c**) The α-solid solution phase (A); (**d**) The I-phase (B).

To further confirm the existence of the I-phase, the ribbons were investigated by TEM, as shown by a bright-field electron micrograph from the central zone of the $Ti_{40}Zr_{40}Ni_{20}$ rod in Figure 3a. The corresponding SAED patterns show the characteristic two-fold and five-fold symmetry of the I-phase at different tilts (shown in Figure 3c,d) corresponding to the X-ray diffraction peaks as indexed (shown in Figure 1a); this demonstrates that the samples maintained a quasicrystal structure. The TEM micrograph of the $Ti_{40}Zr_{40}Ni_{20}$ ribbons, as shown in Figure 3b, reveals the formation of the I-phase grains with a chemical composition of $Ti_{41\pm0.1}Zr_{26\pm0.2}Ni_{32\pm0.3}$ ranging from 64–184 nm. The representative SAED pattern of the I-phase reveals more distorted two-fold and five-fold symmetry in Figure 3e,f. Significantly, the grain size of the ribbons is smaller than that of the rod. This is due to the faster cooling of the single-roll ribbon, its greater nucleation rate, and its smaller grain. Totally, no O element was observed in the EDS patterns, indicating that the probability of the bulk oxidation of the rod sample and ribbons was very slight, and further confirming that the vacuum used to produce the sample was sufficient. The alloy composition was found to be close to the nominal stoichiometry taken. The EDS analysis results of the I-phase (C) of the $Ti_{40}Z_{r40}Ni_{20}$ ribbons in Figure 3b are shown in Figure 3g.

Via image processing using Image-Pro Plus, we were able to quantificationally determine the relative volume fraction of the phases. Although the rod alloy was not a single IQC phase, the IQC formed more than 84.5% of the volume, whereas the other phases formed nearly 15.5%, Confirming that the two alloys fabricated by blow-casting and rapid solidification were both quasicrystal alloys.

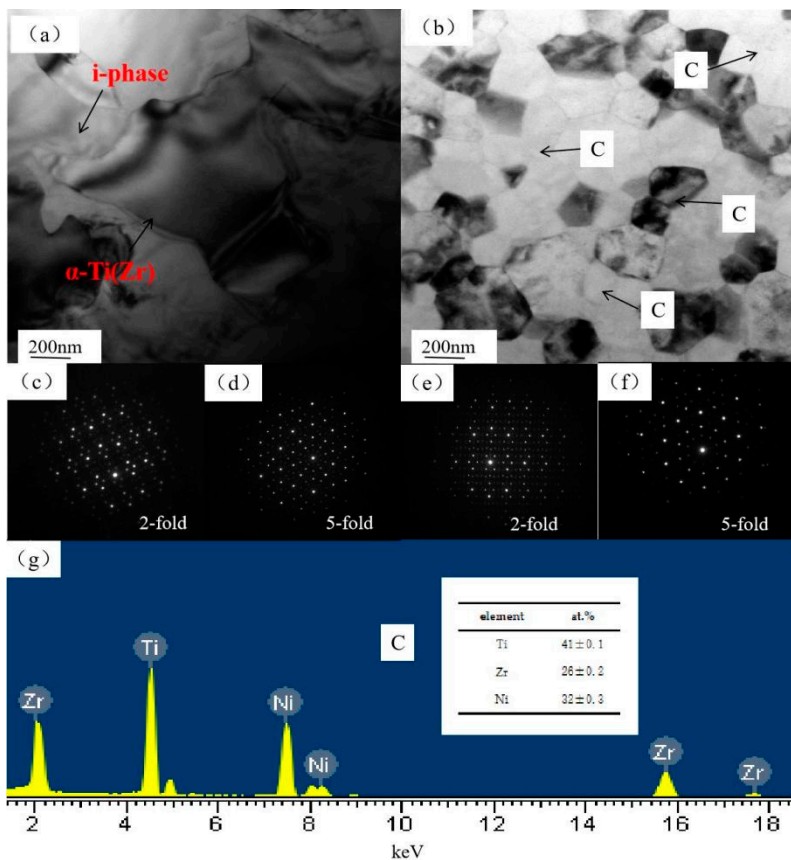

**Figure 3.** (**a**) TEM image, corresponding to (**c**) two-fold, and (**d**) five-fold of the I-phase of the Ti$_{40}$Zr$_{40}$Ni$_{20}$ rod; (**b**) TEM image, corresponding to (**e**) two-fold, (**f**) five-fold, and (**g**) EDS spectrum of the I-phase (C) of the Ti$_{40}$Zr$_{40}$Ni$_{20}$ ribbons.

## 3.2. DSC Analysis

Subsequently, thermal analysis tests were carried out on the Ti$_{40}$Zr$_{40}$Ni$_{20}$ rod and ribbons. Figure 4 exhibits the DSC curve of two specimens with a heating rate of 10 K/min. It can be seen from the DSC curve that there are three endothermic events (867 K, 950 K and 1109 K) of the rod specimen shown in Figure 4a. According to the phase diagram [30] of Ti-Zr-Ni and Qiang et al. [31], during the continuous heating process of the T$_{40}$Zr$_{40}$Ni$_{20}$ rod an isomorphic transition between α-solid solution and β-solid solution occurs first near 867 K, then the I-phase begins to decompose into the C14 phase and the β-solid solution at 950 K. Eventually, near 1109 K, the whole tissue is melted to produce a liquid phase. Based on XRD data, the rod is composed of the I-phase and α-solid solution phase at room temperature. Two endothermic peaks occurred (867 K, 950 K) during the DSC testing process, which is consistent with the two phases calibrated in the XRD pattern.

In order to further confirm the formation of the C14 phase, the T$_{40}$Zr$_{40}$Ni$_{20}$ rod was vacuum annealed at 1020 K for 2 h, followed by furnace cooling. The X-ray diffraction spectrum of the annealed sample is shown in Figure 5a; the quasicrystalline phase disappears, the peak strength of the α-solid solution phase increases, and the C14 phase appears. The results of the backscattered electron image and energy spectrum analysis of the annealed tissue are shown in Figure 5b. The gray phase (D) is the C14 phase (Ti$_{39\pm0.2}$Zr$_{33\pm0.4}$Ni$_{27\pm0.3}$), with the MgZn$_2$ structure type [30,32]. The phase transition near each temperature is shown in Table 1.

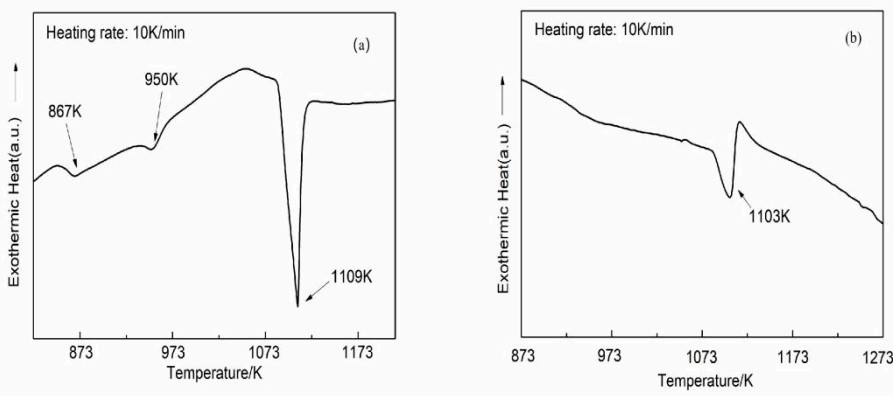

**Figure 4.** The DSC heating trace: (**a**) The Ti$_{40}$Zr$_{40}$Ni$_{20}$ rod; (**b**) The Ti$_{40}$Zr$_{40}$Ni$_{20}$ ribbons.

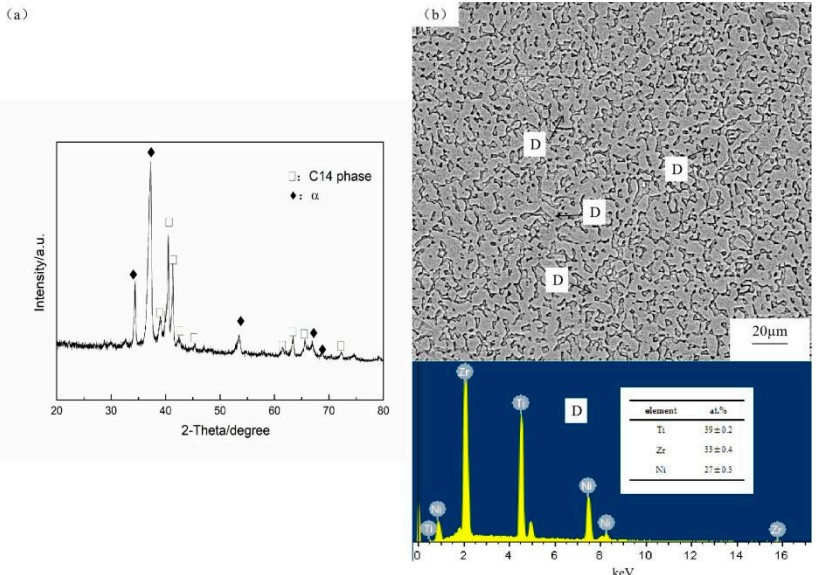

**Figure 5.** (**a**) The XRD patterns; (**b**) The SEM backscattered electron image of the Ti$_{40}$Zr$_{40}$Ni$_{20}$ rod annealed at 1020 K.

**Table 1.** The phase transition process corresponding to each heat absorption peak.

| T(K) | Sample | |
|---|---|---|
| | Rod | Ribbons |
| 867 | α→β | - |
| 950 | I→β + C14 phase | - |
| 1109 | β + C14 phase→Liquid | I→Liquid |

For T$_{40}$Zr$_{40}$Ni$_{20}$ ribbons, a sharp endothermic event is observed between 1073–1123 K on the DSC curve of the specimens in Figure 4b. The I-phase is generated directly from the liquid at around 1103 K, bypassing the C14 phase and β-solid solution. Therefore, the formation mechanism of the quasicrystal phase is different in the two solidification modes, with rapid solidification able to obtain more thermodynamically stable quasicrystalline alloy than blow casting.

### 3.3. Nanoindentation Study

The representative displacement(h)-load(P) for the two materials was studied. The results are given in Figure 6a; the curves of the rod alloy at different peak loads (1.5 and 10 mN) are curves 1, 2 and 3, respectively. The hardness values of nanoindentation tested

are respectively 2.3, 2.26 and 2.0 GPa. The residual indentation depth from 96 to 348 nm can be clearly seen, but hardness decreases from 2.3 GPa to 2 GPa, which is due to a size effect in the sample during the nanoindentation process, such that the H drops with the rise of indentation depth [33].

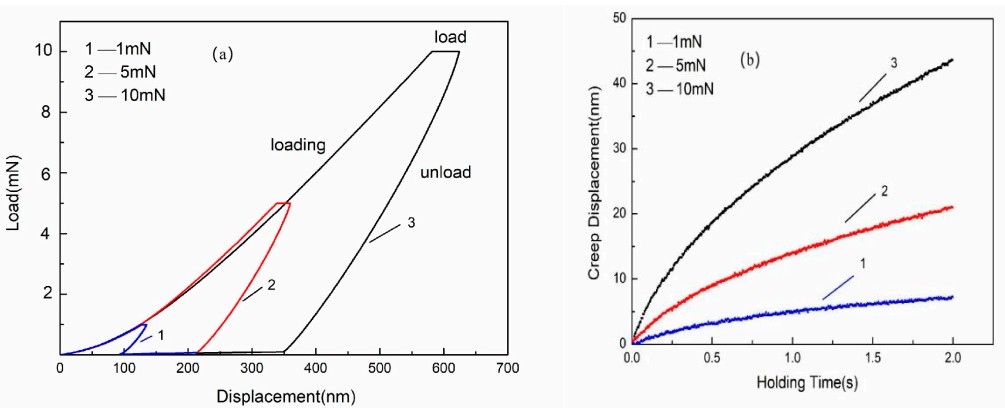

**Figure 6.** (**a**) The representative displacement(h)-load(P); (**b**) The creep displacement curves of the $Ti_{40}Zr_{40}Ni_{20}$ rod at diverse peak load, as measured by Bruker Hysitron TI Premier system.

It also can be observed that there is a creep platform in holding time, and that the elastic recovery phenomenon occurs during the unloading process (shown in the unloading curve) on each P-h curve. This phenomenon indicates it can undergo creep deformation at room temperature. The creep displacement curves of the rod specimen were obtained at different peak loads during the holding load period (in Figure 6b). It can be seen that the nanoindent creep process has transient creep in the early stage of holding time, as well as steady-state creep. Furthermore, creep displacement increases with the rise of peak load, and ranges from 8 nm (appears at 1 mN) to 43 nm (appears at 10 mN). The nanoindentation creep behaviour of the test alloys corresponds to the results of Li and Zheng et al., indicating that creep displacement of the alloy is closely related to the peak load [33,34].

The P-h curves for the two materials with a load limit of 10 mN are also shown in Figure 7. The residual indentation depth is around 350 nm for the rod alloy and 176 nm for the ribbons, respectively. The hardness value of nanoindentation tested was 2.0 GPa (rod), less than 6.2 GPa (ribbons). The hardness was similar to that of Ti(Zr)-based amorphous alloy [35,36], which is 1.5 times the hardness value of common Ti alloy [37]. In comparing the hardness value with micron-sized quasicrystals in other systems, we find that the average hardness value is less than the reported hardness (9 GPa) in Al-based QCs [38]. In addition, the experimental value of the elastic modulus of the rod sample was 45 GPa, which is slightly higher than the 43 GPa and the 21.4 GPa reported by Qiang and Zhao [39,40]. Moreover, there are few studies on the elastic modulus of such ribbons; the value measured in the nanoindentation experiment was 76 GPa. Compared with the Ti40Zr40Ni20 rod, the elastic modulus of the ribbons increased by 68.9%. The elastic modulus of a material is affected by the bond strength between atoms, ions or molecules, crystal structure and micro-organization, etc. [41]. The micro-organization of the rod and ribbons were obviously different; there was no solid solution, and the content of the quasicrystalline phase increased in the ribbons. This not only affects the bonding strength of the overall material but also leads to changes in the local crystal structure and free volume in the alloy [42–45]. The above factors may be responsible for the differences in elastic modulus. This indicates that the increase in the I-phase is not detrimental to improving the elastic modulus of the alloy.

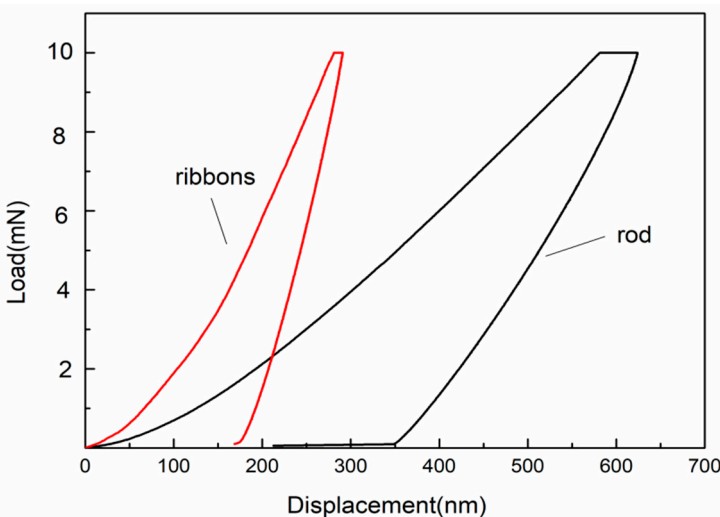

**Figure 7.** The representative displacement(h)-load(P) for the $Ti_{40}Zr_{40}Ni_{20}$ rod and ribbons.

## 4. Conclusions

In summary, we successfully synthesized quasicrystalline $Ti_{40}Zr_{40}Ni_{20}$ alloy via the common blow casting and rapid solidification technique, where the I-phase is the dominant phase. It was found that the product in the $Ti_{40}Zr_{40}Ni_{20}$ quasicrystal rod mainly contained 84.5% icosahedral quasicrystal, as well as 15.5% α-Ti(Zr) phase.

The $Ti_{40}Zr_{40}Ni_{20}$ quasicrystal ribbons were mainly composed of a single I-phase, with the grain reaching 64–184 nm. Additionally, the solidification mechanisms and decomposition temperatures of the quasicrystals were different during two preparation processes. The quasicrystal phase of the $Ti_{40}Zr_{40}Ni_{20}$ rod was generated by the peritectoid reaction of the C14 phase and β-solid solution, while in the $Ti_{40}Zr_{40}Ni_{20}$ ribbons the I-phase was generated directly from the liquid; the decomposition temperatures of the rod and ribbon quasicrystals were 950 K and 1109 K. respectively. This shows that the rapid solidification method can obtain a more thermodynamically stable quasicrystal alloy. The mean room temperature hardness for the quasicrystalline ribbons was around 6.2 GPa, which is slightly greater than the rod (2 GPa) in 10 mN. The mean elastic modulus of the ribbons was about 76 GPa, whereas that of the rod was about 45 GPa. Compared with the rod alloy, the hardness of the ribbons was more than twice as high, and the elastic modulus increased by 68.9%.

**Author Contributions:** Data analysis, J.H., Z.Y., Y.G., J.L.; manuscript writing, J.H.; literature search, J.H., Y.F., H.D., Y.G., J.L.; figures, J.H., Y.F., H.D.; research design, Z.Y., Y.G., J.L.; data interpretation, Z.Y., Y.G., J.L.; data collection, Y.F., H.D. All authors have read and agreed to the published version of the manuscript.

**Funding:** This research was supported by Shaanxi Provincial Key Research and Development Project (Grant2019ZDLGY05-09); Shaanxi Provincial Education Department to Serve the Local Special Plan Project (Grant19JC022); Yulin Science and Technology Bureau Project (Grant2019-121).

**Institutional Review Board Statement:** Not applicable.

**Informed Consent Statement:** Not applicable.

**Data Availability Statement:** Data is contained within the article.

**Acknowledgments:** The authors acknowledge to all the authors who contributed to this article and the teachers who provided the test analysis.

**Conflicts of Interest:** The authors declare no conflict of interest.

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
