# Peer review of "Microstructure and Nanoindentation Behavior of Ti40Zr40Ni20 Quasicrystal Alloy by Casting and Rapid Solidification"

_metals, doi:10.3390/met11101563_

Round 1
Reviewer 1 Report
The study: “Microstructure and Nanoindentation Behavior of Ti40Zr40Ni20 2 Quasicrystal Alloy by Casting and Rapid Solidification” is interesting research in the field of quasicrystalline materials. The results are logical. The draw conclusions are well supported by the obtained results. The paper can be published in Metals after a proper revision.
Comments:
- Can authors quantitatively compare the cooling rate for both crystallization methods?
- The SEM-EDS and TEM-EDS spectrums for figure 2 and 3, respectively, are required.
- Line 93, 107: What is Piont A/B/С, maybe Point?
- Line 114-115: “The EDS analysis results of each point in Figure 3a and Figure 4b are listed in Table 1.” The references are not correct. Figure 3a should be replaced to Figure 2a, and Figure 4b to Figure 3b.
- In figure 3 the font of the scale text should be increased.
- Lines 127-130: “It can be seen from the DSC curve that there are three endothermic event(867K,950K and 1109K ) of rod spencimen shown in Figure 4a, while upon DSC of ribbon spencimens, a sharp endothermic event is observed between 1073K-1123K in Figure 4b. This phenonmeon in DSC curve corresponds to the X-ray diffraction patterns.” This is not clear to which X-ray diffraction patterns the mentioned phenomenon corresponds. In present study the X-ray data was obtained at room temperature, whereas the DSC analysis was performed in a range of temperatures. Authors should discuss this point more precisely or provide a corresponding reference to literature.
- The authors presented the sequences of phase transformations based only on the DSC analysis (Table 2). How did authors concluded that exactly C14-phase was formed? What is the crystalline type of this phase? This part (3.2 DSC analysis) should be strengthened by structure investigations or at least the references to a proper literature are expected.
- Multiple technical mistakes appear through the text:
The sentence (Line 29-33) is too long and must be modified.
The authors use terms i-phase and I-phase through the text. I suppose I-phase is more appropriate
Lines 40: “to be” is correct
Lines 42-43: ceRtainly but not ceNtainly
Lines 56-57: “rotations per minute” is better
Line 102: not Figure 3 but Figure 3a
Line 107: extra bracket
Lines 128, 129: specimens but not speNsimens
Line 159: Li and Zheng
etc
Author Response
Dear reviewer:
I am very grateful to your comments for the manuscript and your very encouraging comments on the merits. According with your advice, we amended the relevant part in manuscript. The revised manuscript is uploaded in the attachment for your viewing.
Thank you very much for your effective suggestions in the article, and we have made up for some of the small experiments and documentary proof to make this article better understood by the readers. If there is something in the article to continue improving, please give us an opportunity to revise again.
Some of your questions were answered below.
Answers of comments:
- Can authors quantitatively compare the cooling rate for both crystallization methods?
Answer:
Other members of our research group have measured the average cooling rate of different cooling methods with temperature recorder(KLCOM100). Based on the research group's accumulated data and literature, the following conclusions are obtained. In the process of casting, the cooling rate at the bottom of the copper mold exceeds 103K/s; the cooling speed can reach 104~106K/s in the process of strip rejection experiment.(2.2 Copper Blow Casting,2.3 Rapid Quenchin)
- The SEM-EDS and TEM-EDS spectrums for figure 2 and 3, respectively, are required.
Answer: The SEM-EDS and TEM-EDS spectrums are given in Figure2(c),(d) and
Figure 3 (g). (Line 124,149)
- Line 93, 107: What is Piont A/B/С, maybe Point?
Answer:
In order not to cause misunderstanding, appropriate modifications have been made here. The chemical composition list collected by the energy spectrum has been deleted and inserted into the spectral picture in the form of illustrations.
A and B represent the two phases (α-solid solution phase and I-phase) in rod, point C represent I-phase in ribbons. Chemical composition of each phase in Figure 2(a) and Figure 3(b) was measured four times and the average values are shown in the interpolation table in Figure 2(c), (d) and Figure 3 (g). (3.1. Structure Characterization and Morphology Line 115,134)
- Line 114-115: “The EDS analysis results of each point in Figure 3a and Figure 4b are listed in Table 1.” The references are not correct. Figure 3a should be replaced to Figure 2a, and Figure 4b to Figure 3b.
Answer: The sentence has been modified. The EDS analysis results of each point in Figure 2a and Figure 3b are shown in the interpolation table in Figure 2(c),(d) and Figure 3 (g). (Line 124,149)
- In figure 3 the font of the scale text should be increased.
Answer: The font in the diagram has been modified appropriately in Figure 3. (Line 149)
- Lines 127-130: “It can be seen from the DSC curve that there are three endothermic event(867K,950K and 1109K ) of rod spencimen shown in Figure 4a, while upon DSC of ribbon spencimens, a sharp endothermic event is observed between 1073K-1123K in Figure 4b. This phenonmeon in DSC curve corresponds to the X-ray diffraction patterns.” This is not clear to which X-ray diffraction patterns the mentioned phenomenon corresponds. In present study the X-ray data was obtained at room temperature, whereas the DSC analysis was performed in a range of temperatures. Authors should discuss this point more precisely or provide a corresponding reference to literature.
Answer:
I'm really sorry that you misunderstood the original meaning of this sentence due to my lack of language expression ability and knowledge.
“This phenonmeon in DSC curve corresponds to the X-ray diffraction patterns” This sentence originally intended to express that the two melting peaks(867K, 950K) of the rod exactly correspond to the two phases calibrated in the XRD pattern. In detail, based on XRD patterns, the rod is composed of α-solid solution phase and I-phase at room temperature. According to the phase diagram and literature, it is also known that the α-solid solution phase transforms at 873K and the quasicrystal phase dissolves around 950K, which corresponds to the first two melting peaks in DSC test. This phenomenon is also consistent with the two phases calibrated in the XRD. I have remade detailed changes (3.2 DSC Analysis) in order to stop causing misunderstanding. (3.2. DSC Analysis Line 153-163)
- The authors presented the sequences of phase transformations based only on the DSC analysis (Table 2). How did authors concluded that exactly C14-phase was formed? What is the crystalline type of this phase? This part (3.2 DSC analysis) should be strengthened by structure investigations or at least the references to a proper literature are expected.
Answer:
I am sorry for my omission in this aspect, to which I have added part of the proof literature and a small number of experiments to prove the generation of the C14 phase (3.2 DSC Analysis).
The rod was cooled with the furnace after 2h of annealing experiments at an annealing temperature of 1020K. The tissue of the annealed samples was then observed, and the XRD characterization was performed. Energy spectrum analysis and object phase results jointly demonstrate the generation of the C14 phase. (3.2. DSC Analysis Line 164-170)
Answers of Multiple technical mistakes appear through the text:
- The sentence (Line 29-33) is too long and must be modified.
The length of the sentence has been corrected.(Line 28-31,32-33)
- The authors use terms i-phase and I-phase through the text. I suppose I-phase is more appropriate
The i-phase in the article has been replaced with I-phase.
- Lines 40: “to be” is correct
The”been” has been corrected to “to be”. (Line 41)
- Lines 42-43: ceRtainly but not ceNtainly
The spelling error has been corrected.(Line 43)
- Lines 56-57: “rotations per minute” is better
The sentence statement has been modified to read:”on a single copper roller that rotations 2500 per minute” (2.3 Rapid Quenching)
- Line 102: not Figure 3 but Figure 3a
Correct the image annotation error. (Line 129)
- Line 107: extra bracket
The error has been corrected. (Line 134)
- Lines 128, 129: specimens but not speNsimens
The spelling error has been corrected.(Line 154,155)
- Line 159: Li and Zhengetc
The spelling error has been corrected. (Line 201)
Thank you.
JunLi Hou
Come from Xi’an Technological University

Reviewer 2 Report
I would like to acknowledge authors for the manuscript “Microstructure and nanoindentation behavior of Ti40Zr40Ni20 quasicrystal alloy by casting and rapid solidification. Unfortunately, I cannot recommend this manuscript for publication. My main concern is that the influence of the microstructure on the elastic properties of rods and ribbons was not considered. The difference in the elastic properties may not be related to the presence or absence of alpha- and i-phases, but may be related to a completely different microstructure only, which is very different for rods and ribbons reported in this manuscript. It seems that this research is not completed.
Other major corrections:
1) “2. Experimental procedure”: please, specify all steps of sample preparation, including temperatures and duration of each step. Was X-ray diffraction of starting materials (used for synthesis) performed too? Where starting materials were acquired from? What is their crystallinity? Not enough details to reproduce the sample preparation.
2) Figure 2: was chemical composition measured only for 2 points in the rod? No statistics? Where is chemical composition of ribbons?
3) “3.2. DSC analysis”: have authors performed high-temperature x-ray diffraction measurements to support their conclusions? Seems to be too much discussion which is not based on experimental results reported in this manuscript.
Minor corrections:
1) Abstract, line 15: “hardnessand” is two words, space between them is missing.
2) Could authors check, that none of “i-phase” was replaced by automatic spelling check by “I-phase” in the middle of sentence?
3) Introduction, line 40 “is hard to been fabricated” replace by “is hard to be fabricated”
4) Introduction, line 43, “While, “ – sentence does not requires “while” at the beginning of it.
5) “3.1 Structure characterisation and morphology”, line 74. Paragraph should not start with words “Figure X shows…” – it is a style of undergraduate report. This comment is relevant for all other paragraphs too.
6) Figure 2 – check label, it is misleading. What are (e) and (f), if (c) and (d) are for both, rods and ribbons?
Author Response
Dear reviewer:
I am very grateful to your comments for the manuscript. According with your advice, we amended the relevant part in manuscript. Some of your questions were answered below. The revised manuscript is uploaded in the attachment for your viewing.
Thank you very much for your suggestions, which are very necessary. We understand your concerns about this issue. Indeed, it is well known that the elastic modulus related to the bonding mode, crystal structure, chemical composition, microorganization, temperature and other factors are also affected by the alloy composition and processes. However, the implication is that the performance is affected by the organization, which is composed of different phases. The structure of different phases and the bonding mode of the atoms forming the phase will affect the material properties. I am very sorry that our expression has caused you a misunderstanding. Of course we have renewed the discussion of this area to facilitate a better understanding.
If there is something in the article to continue improving, please give us an opportunity to revise again.
The answer of other major corrections:
- “2. Experimental procedure”: please, specify all steps of sample preparation, including temperatures and duration of each step. Was X-ray diffraction of starting materials (used for synthesis) performed too? Where starting materials were acquired from? What is their crystallinity? Not enough details to reproduce the sample preparation.
Answer:
I apologize for the lack of a detailed description of the experimental aspect. I also appreciate your advice. I have filled in this missing part(such as, the equipment current and other parameters in the experiment) in the manuscript. (Line 51-80) In addition, due to the size of raw materials (Ti, Zr, Ni), it is difficult to collect XRD data and crystallinity data. (diameter 0.1mm, length 0.5mm round rod)
- Figure 2: was chemical composition measured only for 2 points in the rod? No statistics? Where is chemical composition of ribbons?
Answer:
In order not to cause misunderstanding, appropriate modifications have been made here. The chemical composition list collected by the energy spectrum has been deleted and inserted into the spectral picture in the form of illustrations.
A and B represent the two phases (α-solid solution phase and I-phase)in rod, point C represent I-phase in ribbons. Chemical composition of each phase in Figure 2(a) and Figure 3(b) was measured four times and chemical composition is numerically similar. So, the average values are listed in the interpolation table in Figure 2(c),(d). Due to the thinness of SEM pictures of ribbons. Chemical composition of ribbons is obtained by TEM-EDS(in Figure 3(b)) and is also listed in the interpolation table in Figure 3(g). (Line 124,149)
- “3.2. DSC analysis”: have authors performed high-temperature x-ray diffraction measurements to support their conclusions? Seems to be too much discussion which is not based on experimental results reported in this manuscript.
Answer:
I'm really sorry that you misunderstood the original meaning of this sentence due to my lack of language expression ability and knowledge.
“This phenonmeon in DSC curve corresponds to the X-ray diffraction patterns” This sentence originally intended to express that the two melting peaks(867K,950K) of the rod exactly correspond to the two phases calibrated in the XRD pattern. In detail, based on XRD patterns, the rod is composed of α-solid solution phase and I-phase at room temperature. According to the phase diagram and literature, it is also known that the α-solid solution phase transforms at 873K and the quasicrystal phase dissolves around 950K, which corresponds to the first two melting peaks in DSC test. This phenomenon is also consistent with the two phases calibrated in the XRD. I have remade detailed changes(3.2 DSC Analysis) in order to stop causing misunderstanding. (Line 153,163)
Answers of minor corrections:
- Abstract, line 15: “hardnessand” is two words, space between them is missing.
-Abstract: whitespace has been added to “hardnessand”. (Line 15)
- Could authors check, that none of “i-phase” was replaced by automatic spelling check by “I-phase” in the middle of sentence?
-The“i-phase” in the middle of sentence has been changed to“I-phase”.
- Introduction, line 40 “is hard to been fabricated” replace by “is hard to be fabricated”
-The grammatical error has been corrected at Introduction. (Line 41)
- Introduction, line 43, “While, “ – sentence does not requires “while” at the beginning of it.
-The “While”at the beginning of the sentence has been removed.
- “3.1 Structure characterisation and morphology”, line 74. Paragraph should not start with words “Figure X shows…” – it is a style of undergraduate report. This comment is relevant for all other paragraphs too.
-The beginning of the paragraph was appropriately revised
- Figure 2 – check label, it is misleading. What are (e) and (f), if (c) and (d) are for both, rods and ribbons?
-Sorry that we lost the note to the icon. We have modified this in the annotations in Figure 3. (c) and (d) correspond to the 2-fold and 5-fold of quasicrystals in the rod, (e) and (f) correspond to the 2-fold and 5-fold of quasicrystals in the ribbons. I have modified this in the annotations in Figure 3. (Line151)
We hope you are satisfied with the revised version, however, if there is more question, we are willing to revise it again.
Thank you.
JunLi Hou
Come from Xi’an Technological University

Reviewer 3 Report
In their contribution "Microstructure and Nanoindentation Behavior of Ti40Zr40Ni20 Quasicrystal Alloy by Casting and Rapid Solidification", Hou et al. report on the syntheses, structural characterizations, the hardness, and the elastic modulus of the aforementioned material. To do so, the authors made use of different (state-of-the-art) experimental techniques. The work is well suited for Metals; however, there are several issues, which need to be revised prior to a publication of this contribution:
- Introduction: with regard to the structures and properties of quasicrystals, there are also quite a number of reviews, which have been published in Chem. Soc. Rev. and could be cited by the authors.
- It could be helpful for the readers, if the authors include a representation showing the structure of the inspected material.
- In the synthesis section, it is mentioned that the compounds were obtained under a dry argon atmosphere - how large were the contents of air and moisture within that argon atmosphere? Please include this relevant information in the manuscript.
- The authors collected EDS data in order to confirm the compositions of the obtained materials; however, the authors did not show the corresponding spectra. Please include this relevant data in the manuscript.
- Figure 3: what is shown in the insets (e) and (f)?
- Page 4: " This phenonmeon in DSC curve corresponds to the X-ray diffraction patterns." How can the phenomenon observed for the DSC measurements be related to the outcome of the XRD measurements, if no temperature-dependent XRD data have been collected by the authors? Please revise this statement. If the authors refer to a research report that has been previously published elsewhere, please include an appropriate reference to cite this research.
- Page 6: "The hardess similar to that ofTi (Zr)-based amorphous alloy, which is 1.5 times the hardness value of ordinary Ti alloy." Please include an appropriate reference to prove this statement.
- There are several typing errors, which should be corrected (see e.g. p. 1: "... at of the rod,and the..." or p. 2.: "... From these ingots,The ...").
Author Response
Dear reviewer:
I am very grateful to your comments for the manuscript and your very encouraging comments on the merits. According with your advice, we amended the relevant part in manuscript. The revised manuscript is uploaded in the attachment for your viewing.
The literature you recommend in the structure and properties of the introduction part gives us great help, and the review of the literature in Chem. Soc. Rev., which gives us a further understanding of the structure of the aligned crystal, thank again.If there is something in the article to continue improving, please give us an opportunity to revise again.If there is something in the article to continue improving, please give us an opportunity to revise again.
Some of your questions were answered below.
1) Introduction: with regard to the structures and properties of quasicrystals, there are also quite a number of reviews, which have been published in Chem. Soc. Rev. and could be cited by the authors.
Answer:
We cite two articles in Chem. Soc. Rev. to complement a better understanding of quasamystal structure and performance. (Line 23)
- Tsai, A.P.Discovery of stable icosahedral quasicrystals: progress in understanding structure and properties. Chemical Society reviews,2013,42,5351-5365; doi:10.1039/c3cs35388e.
- Dubois, J.Properties and applications of quasicrystals and complex metallic alloys. J. Chemical Society reviews,2012,41,6760-6077; doi:10.1039/c2cs35110b
2)It could be helpful for the readers, if the authors include a representation showing the structure of the inspected material.
Answer:
Since the structure of quasicstalline icosahedrdral is difficult to understand, we cite three literature in Chem. Soc. Rev. . There are better graphical simulations in these literature allowing readers to understand quasicystalline. Many thanks to the authors of these articles. (Line 22,2
- Tsai, A.P. Discovery of stable icosahedral quasicrystals: progress in understanding structure and J. Chemical Society reviews,2013,42,5351-5365; doi:10.1039/c3cs35388e.
- Dubois, J. Properties and applications of quasicrystals and complex metallic alloys. J. Chemical Society reviews,2012,41,6760-6077; doi:10.1039/c2cs35110b
- Abe, Electron microscopy of quasicrystals -where are the atoms? J. Chemical Society reviews,2012,41,6778-6786; doi:10.1039/c2cs35212e.
3) In the synthesis section, it is mentioned that the compounds were obtained under a dry argon atmosphere - how large were the contents of air and moisture within that argon atmosphere? Please include this relevant information in the manuscript.
Answer:
During the experiment, the equipment was vacuumized to 10×10-3Pa and then filled with argon gas, and the final pressure gauge showed 0.07mpa. At this time, according to the gas state equation, the proportion of argon in the equipment is 0.8996, and that of other gases is 0.1004. It has been shown in the text. (2. Experimental Procedure).
4) The authors collected EDS data in order to confirm the compositions of the obtained materials; however, the authors did not show the corresponding spectra. Please include this relevant data in the manuscript.
Answer:
Spectra have been supplemented. The SEM-EDS and TEM-EDS spectrums are given in Figure2(c),(d) and Figure3(g). (Line 124,149)
5) Figure 3: what is shown in the insets (e) and (f)?
Answer:
Sorry that we lost the note to the icon. We have modified this in the annotations in Figure 3. (c) and (d) correspond to the 2-fold and 5-fold of quasicrystals in the rod, (e) and (f) correspond to the 2-fold and 5-fold of quasicrystals in the ribbons. (Line 151)
6)Page 4: " This phenonmeon in DSC curve corresponds to the X-ray diffraction patterns." How can the phenomenon observed for the DSC measurements be related to the outcome of the XRD measurements, if no temperature-dependent XRD data have been collected by the authors? Please revise this statement. If the authors refer to a research report that has been previously published elsewhere, please include an appropriate reference to cite this research.
Answer:
We are really sorry that you misunderstood the original meaning of this sentence due to my lack of language expression ability and knowledge.
“This phenonmeon in DSC curve corresponds to the X-ray diffraction patterns” This sentence originally intended to express that the two melting peaks(867K,950K) of the rod exactly correspond to the two phases calibrated in the XRD pattern. In detail, based on XRD patterns, the rod is composed of α-solid solution phase and I-phase at room temperature. According to the phase diagram and literature, it is also known that the α-solid solution phase transforms at 873K and the quasicrystal phase dissolves around 950K, which corresponds to the first two melting peaks in DSC test. This phenomenon is also consistent with the two phases calibrated in the XRD. I have remade detailed changes(3.2 DSC Analysis) in order to stop causing misunderstanding.(Line 153-163)
7)Page 6: "The hardess similar to that of Ti (Zr)-based amorphous alloy, which is 1.5 times the hardness value of ordinary Ti alloy." Please include an appropriate reference to prove this statement.
Answer:
I am very sorry for my negligent omission of the supporting literature on this respect, and we have made additions.Supporting references are listed in the bibliography. (Line 211)
- Amiya, K.;Nishiyama, ;Inoue, N.;Masumoto, T. Mechanical strength and thermal stability of Ti-based amorphous alloys with large glass-forming ability. J. Materials Science and Engineering,1994,179-180,692-696.
- Shen,X.J. Study on the Structure and Mechanical Properties of Metal glass Reinforced Ti Alloy Composite by 3D printing. D. Huazhong University of Science & Technology,2019.
- Yao,X.J.;Wang, K.;Gao,L.;Zhang, X.B.;Liu, C. Microstructure and properties of TC16 titanium alloycold rolling external thread. J.Scientific and Technological Innovation(In Chin.),2021,26,38-41.
8) There are several typing errors, which should be corrected (see e.g. p. 1: "... at of the rod,and the..." or p. 2.: "... From these ingots,The ...").
Answer: The spelling error was corrected at p1 and P2.
We hope you are satisfied with the revised version, however, if there is more question, we are willing to revise it again.
Thank you.
JunLi Hou
come from Xi’an Technological University

Round 2
Reviewer 1 Report
The reviewer is satidfied by the answers. The paper can be published in present state.
Author Response
Dear reviewer:
Thank you again for your advice and encouragement.
With best wishes, thank you.
JunLi Hou
Come from Xi’an Technological University